# Neurologic Music Therapy in Geriatric Rehabilitation: A Systematic Review

**DOI:** 10.3390/healthcare10112187

**Published:** 2022-10-31

**Authors:** Jagoda Rusowicz, Joanna Szczepańska-Gieracha, Paweł Kiper

**Affiliations:** 1Department of Physiotherapy, Wroclaw University of Health and Sport Sciences, 51-612 Wrocław, Poland; 2Healthcare Innovation Technology Lab, IRCCS San Camillo Hospital, 30126 Venezia, Italy

**Keywords:** neurorehabilitation, neurological diseases, Parkinson’s disease, stroke

## Abstract

(1) Introduction: Neurologic music therapy (NMT) is a non-pharmacological approach of interaction through the therapeutic use of music in motor, sensory and cognitive dysfunctions caused by damage or diseases of the nervous system. (2) Objective: This study aimed to critically appraise the available literature on the application of particular NMT techniques in the rehabilitation of geriatric disorders. (3) Methods: PubMed, ScienceDirect and EBSCOhost databases were searched. We considered randomized controlled trials (RCTs) from the last 12 years using at least one of the NMT techniques from the sensorimotor, speech/language and cognitive domains in the therapy of patients over 60 years old and with psychogeriatric disorders. (4) Results: Of the 255 articles, 8 met the inclusion criteria. All papers in the final phase concerned the use of rhythmic auditory stimulation (RAS) (sensorimotor technique) in the rehabilitation of both Parkinson’s disease (PD) patients (six studies) and stroke patients (SPs) (two studies). (5) Conclusion: All reports suggest that the RAS technique has a significant effect on the improvement of gait parameters and the balance of PD patients and SPs, as well as the risk of falls in PD patients.

## 1. Introduction

The ageing of the population is a long-term phenomenon that has been visible in Europe for several decades. The increasing number and participation of older people in society not only creates a number of opportunities but also new challenges, especially related to public health [1].

Psychopathology in geriatrics should be considered by taking into account the groups of causes of mental disorders in old age to which they belong: Psychological and social factors related to old age, Ageing of the body along with neurodegenerative processes, which are manifested by psycho-organic disorders, Somatic diseases and their treatment, The course of mental disorders that started in earlier stages of life and changes in their psychological image [2].

NMT is a non-pharmacological method of interaction through the therapeutic use of music in motor, sensory and cognitive dysfunctions caused by damage or diseases of the nervous system. The supporters of this approach treat music as a stimulus that directly affects the neurophysiological processes of the brain. The use of music in neurorehabilitation is based on neurophysiological theories and research on the influence of music production and perception on cognitive processes and principles of learning through movement [3,4,5].

NMT consists of 20 standardized and clinically applied techniques that rehabilitate three areas: sensorimotor, speech/language and cognitive (Table 1) [6]. Therapeutic activities are aimed at achieving non-musical goals. To optimize this process, NMT uses the Transformation Design Model (TDM) to translate theoretical knowledge into clinical practice. It promotes the effective assessment, design and implementation of therapeutic musical interventions [4,7].

With this systematic review we tried to find answers to the following questions: Can NMT, as a relatively low-cost and non-invasive method, be effective in the rehabilitation of geriatric patients? Which techniques and in which diseases and disorders is NMT successfully applied? In addition, we wanted to locate areas (therapy and rehabilitation of individual conditions) where RCTs are being conducted (providing strong evidence of effectiveness). This study aimed to critically appraise the literature on the application of NMT techniques in the rehabilitation of psychogeriatric disorders.

## 2. Materials and Methods

### 2.1. Search Strategy

We carried out a systematic review of randomized controlled trials (RCTs) on the benefits of NMT in the treatment of psychogeriatric disorders. The verification system was based on PRISMA guidelines and Cochrane Handbook for Systematic Reviews of Interventions to ensure transparent and complete reporting in our study [8,9,10]. We searched the following electronic databases: PubMed, ScienceDirect and EBSCOhost. In the selection of keywords, the names of individual NMT techniques from all three areas were considered [Appendix A]. Data extraction was performed by two researchers and then checked by a third one. Our strategy included three phases. Initially, potentially valuable articles were identified by their titles. Screening of abstracts was performed by two independent reviewers. Disagreements relating to the full text inclusion were solved by a third reviewer. After narrowing the number of articles to those meeting the inclusion criteria, we gained access to full versions of the publications, eventually qualifying the articles for review (Figure 1).

### 2.2. Inclusion Criteria

The articles selected for this literature review fulfilled the following inclusion criteria: published in English in the last 12 years, i.e., from 1 January 2010 to 1 March 2022, and including patients aged 60 years and over who have been diagnosed with geriatric disorders and who have been rehabilitated using at least one of the NMT techniques. The search excluded studies in the form of letters, preliminary findings, literature reviews and case studies. The following information was extracted from each article: year of publication, study group characteristics, therapies used, duration of intervention, questionnaire(s)/tool(s) used to measure and results indicating the degree of efficiency before and after therapy.

### 2.3. Assessment of Risk of Bias 

Risk of bias for the included studies was assessed independently by two reviewers, who were supported by a third researcher in case of disagreement using the Risk of Bias 2 (RoB 2) tool (Figure 2). Assessment was conducted following the criteria stated by the Cochrane Collaboration in the Cochrane Handbook for Systematic Reviews of Interventions [11]. Methodological quality of studies was recorded in the risk of bias table [Appendix A].

## 3. Results

### 3.1. Literature Search

In the first phase of the electronic literature search, 255 potentially relevant publications were found. A total of 170 studies were rejected, because the abstracts did not meet the inclusion criteria. Based on the available summaries, 85 publications were accepted for further full-text analysis. In addition, a manual search of the cited literature within approved articles was carried out. Thus, two additional studies were included. Finally, we included eight articles in this systematic review. The most common reason for rejecting an article in the second phase was the use of music without therapy or a different method of music therapy. The second most common reasons for rejection were too big a difference in the ages of participants and the type of study. The characteristics of the studies qualifying for review and a full list of rejected papers with motivation are included in the Appendix A.

### 3.2. Participants and Study Characteristics

Due to the specificity of the population of elderly people with geriatric disorders in the selected studies, it was not possible to create unambiguously detailed characteristics of the studied groups. However, six of the studies that were qualified for the final stage concerned the use of rhythmic auditory stimulation (RAS) for gait rehabilitation in Parkinson’s disease (PD) patients and two studies concerned RAS in stroke patients (SPs).

### 3.3. RAS in Parkinson’s Disease 

Six out of eight studies were concerned with the use of RAS in the rehabilitation of gait parameters in PD. It is difficult to compare the results because of the different modifications to the classic RAS training. However, the articles focused on evaluating the parameters of gait, balance, and falls. Multimodal balance training supported by RAS can improve balance performance in PD patients with mild cognitive impairment and stage 4 H&Y [12]. Significant improvements in PD gait were obtained after RAS training in parameters (speed; stride length; cadence; right and left ankle dorsiflexion; fall index and fear of falling; reduction in patient step time variability) [13,14,15]. Additionally, finger tapping training led to a significant increase in gait speed (*p* < 0.005) and gait cadence [16].

#### 3.3.1. RAS and Multimodal Balance Training in PD

The study by Capato and colleagues (2020) compared RAS-supported multimodal balance training with regular multimodal training. Results revealed that, immediately after the intervention, both intervention groups improved significantly at the Mini-Best performance level. In both groups, the results remained at the level of 1 month of observation [12]. Outcomes were maintained only in the RAS-supported intervention group after 6 months of follow-up. In this group, improvements were registered in Part 3 of the Movement Disorder Society–Unified Parkinson’s Disease Rating Scale (MDS-UPDRS) and in the Berg Balance Scale (BBS) immediately after the intervention.

#### 3.3.2. RAS and Treadmill Training in IPD

Calabrò and colleagues (2019) assessed the effectiveness of treadmill training in combination with RAS for mobility, balance and gait parameters by correlating electroencephalography (EEG) changes with behavioural changes (gait) in order to determine the alleged neurophysiological basis for gait improvement [15]. Significant improvements were observed in functional gait assessment (*p* < 0.001), gait quality index (*p* < 0.001), Unified Parkinson Disease Rating Scale (*p* = 0.001) and Tinetti Fall Performance Scale (*p* < 0.001) after RAS training.

#### 3.3.3. RAS and Effects of Auditory–Motor Entrainment in IPD

Janzen and colleagues (2019) examined instantaneous effects of auditory–motor entrainment within effector systems [16]. The results implied that finger tapping training was followed by a significant increase in gait speed (*p* < 0.005). Significant changes in gait cadence were found in the finger tapping group (*p* < 0.005) after training, but not after arm swing (*p* = 0.879) and control (*p* = 0.759) training. Summarizing the available results, it can be concluded that auditory–motor entrainment in one effector system can stimulate the other effector system.

#### 3.3.4. RAS and Falls in PD

Another study focused on RAS in fall reduction [13]. For this study, both the experimental and control groups had significant improvements in performance at week 8 from the start of the intervention. At week 16, significant improvements were registered for speed, stride length, cadence, right and left ankle dorsiflexion, fall index and fear of falling in the experimental group. It should be noted that bilateral ankle dorsiflexion correlated significantly with changes in gait, fear of falling and fall rate. This may indicate that ankle dorsiflexion is a potential kinematic mechanism by which RAS may be useful in reducing falls.

#### 3.3.5. Ecological RAS vs. Artificial RAS in PD

The aim of the study by Murgia and colleagues (2018) was to examine if a PD rehabilitation combined with an ecological RAS could be more effective than the same programme integrated with an artificial RAS [17]. The results of the experiment suggest that both groups improved in most biomechanical and clinical measurements, regardless of the type of sound. Exploratory analyses were also performed for the separate groups, which showed improvements in spatiotemporal parameters only for the organic RAS group (*p* = 0.001).

#### 3.3.6. RAS and Sensorimotor Timing Skills in Idiopathic Parkinson’s disease (IPD)

The last article by Bella and colleagues (2017) explored the role of sensorimotor time measurement skills, tested with walk and tap tasks, to assess the effectiveness of RAS in people with IPD [14]. Both gait speed and stride length improved significantly and were maintained one month after music gait training (MCGT; *p* < 0.05). Immediately after training, a significantly reduced variation in patients’ step time was noted (*p* < 0.05). The effect did not persist during the follow-up period (*p* = 0.13). An important aspect of the study is to explore individual factors such as initial gait speed or performance in timed (sensorimotor) tasks that influence the effectiveness of RAS.

### 3.4. RAS in Stroke

RAS also showed benefits for SPs. In the GTBR group, there were significant changes in gait symmetry and in the magnitude of decreases in gait symmetry per step time. The combination of treadmill training and RAS resulted in better functional gait performance (significant changes in speed, cadence and stride length) compared with standard treadmill training [18,19].

#### 3.4.1. Bilateral RAS in SP

The first of two articles by Lee and colleagues (2018) considered the effect of gait training with bilateral RAS (GTBR) on lower limb rehabilitation in SP patients [18]. In the GTBR group, significant changes were obtained in gait symmetry (*p* < 0.05) and in the size of decreases in gait symmetry per step time. Both groups showed improvements in gait ability on velocity and cadence, and scored significantly better (*p* < 0.05) on the BBS, Timed Up and Go (TUG) and Fugl-Meyer assessment (FMA) tests relative to baseline measurements. Summarising the results of both groups, it can be concluded that GTBR allowed greater improvements in gait parameters in SP patients relative to the group receiving standard gait training.

#### 3.4.2. RAS-Treadmill Training in SP

A study by Mainka and colleagues (2018) focused on comparing the effectiveness of a combination of treadmill training (RAS-TT) versus treadmill training (TT) in the rehabilitation of functional gait in stroke [19]. There were significant changes in speed (*p* < 0.001), cadence (*p* = 0.001) and stride length (*p* < 0.001) in the RAS-TT group before and after. There were also significant time effects for these parameters in the FGS test. The RAS-TT group had significantly higher scores compared with the other groups. The researchers used the fact that music allows for the correction of step frequency and improvement of gait pattern, mainly through the action of auditory motor feedback to reduce stress, both motor and psychological. Appendix A [4,20,21,22,23,24,25,26,27,28,29,30,31,32,33,34,35,36,37,38,39,40,41,42,43,44,45,46,47,48,49,50,51,52,53,54,55,56,57,58,59,60,61,62,63,64,65,66,67,68,69,70,71,72,73,74,75,76,77,78,79,80,81,82,83,84,85,86,87,88,89,90,91,92,93,94,95,96] presents the full characteristics of the studies included in the systematic review.

### 3.5. Risk of Bias in Included Studies

Included studies showed some concerns in the risk of bias assessment overall, in which randomization, concealment and missing outcome data were unclear. Two studies were assessed as low risk of bias. One study was rated as high risk of bias due to deviations from the intended intervention, as well as insufficient randomization information and missing results.

## 4. Discussion

The aim of this review was to critically assess the available literature on the application of NMT techniques in the rehabilitation of geriatric disorders. We found eight studies that met the inclusion criteria. All of them were based on the RAS technique (sensorimotor domain) in the movement rehabilitation of PD patients and SPs. RAS aims to develop and maintain a physiological rhythmic motor activity through rhythmic auditory cues. In a series of ground-breaking studies starting in the mid-1990s, it was discovered that this technique permanently improved walking speed, step length and cadence in PD [97,98]. In general, the positive effects of RAS training based on metronome or music are well grounded in terms of the PD literature. Therefore, it is not surprising that our review includes modifications to the RAS technique, combining this technique with multimodal balance training, treadmill training, gait training supported by bilateral RAS and comparison of rehabilitation programme effects integrated with ecological or artificial RAS.

The novelty of comparing RAS-supported multimodal balance training with regular multimodal training study is based on the fact that, for the first time, the importance of specialist physiotherapy with a specific exercise protocol with RAS has been demonstrated, which can improve the balance performance of PD patients in advanced stages of the disease and mild cognitive decline [12]. Researchers report that only a few studies have covered a subgroup of Hoehn and Yahr Scale stage 4 patients, but none of them specifically focused on RAS balance training in advanced stages of the disease. The message is that multimodal balance training (both with and without RAS) is possible in patients in advanced stages of the disease, because it did not cause falls and serious adverse events. The lack of influence on the gait rate results in both groups was interpreted as patients in advanced stages of the disease experience difficulties transferring the balance improvement to the gait task. However, as noted, the use of RAS can improve training effects by making them more pronounced than training without RAS. Reports of the efficacy of multimodal RAS intervention may provide important guidance for practitioners caring for PD patients in advanced stages, as there is still a lack of recommendations for the optimal approach in motor rehabilitation for this group [12,99].

The data presented by Calabrò and colleagues [15] imply that the use of RAS training compared with training without RAS allows for improved overall gait quality and better results in the areas of number and length of gaits and balance. However, the absence of a significant difference in RAS and non-RAS training in terms of improved gait, turning and stride duration may mean that the rehabilitation programme by itself, not cueing, influenced the improvement. Other main conclusion of this study is that only in the group receiving RAS training was frontotemporal connectivity involved in the α frequency range. This functional connectivity is strongly associated with cognitive performance in PD, due to deterioration with cognitive decline. Deterioration of α is a marker of degeneration of the ascending diffuse projection systems that control attention. Therefore, the use of music as an external cue allows for increased levels of attention, and thus improved performance and participation of subjects, as can be seen in the low variability of outcome measures after the RAS intervention.

Interesting conclusions appeared from the study about rhythmic priming across effector systems [16]. The possibility of modulating gait speed through RAS training of arm or finger movements was tested. The first group was asked to tap their fingers in synchrony with a metronome. Participants in the second group were asked to alternate sways synchronised to a metronome. Interestingly, only participants in the first group (finger tapping) achieved significant increases in gait speed and cadence before training. The second group showed no change in gait speed after training. These reports suggest that auditory–motor interaction in one effector system can stimulate the other effector system. Quite surprisingly, the priming effect on gait was only observed in the finger tapping condition. The authors emphasise that the results of the study are relevant for the development of motor rehabilitation. The study is an important step for further exploration of the mechanisms underlying the coupling between effectors, and consequently the development of new therapeutic pathways adapted to the capabilities of PD patients.

Additionally, the use of RAS was effective in reducing falls in PD patients. Discontinuation of the intervention resulted in an increase in falls and gait speed and a reduction in ankle dorsiflexion. Correlational analyses showed a noticeable association between the reduction in falls with improvements in dorsiflexion and gait speed [13]. Overall, the improvement in gait parameters (gait speed, velocity, stride length, and cadence) and balance due to the RAS training was confirmed in all studies. Temporal cues are recognised as important factors that reinforce the underlying physiology of temporal pattern formation in the basal ganglia. As a result, they enhance motor learning. RAS as a coordinative sensory input unifies the temporal functions of the basal ganglia loops and also to increase gait velocity and stride length [100]. It is worth noting that the low cost of therapy and the ability to safely implement training even in a home setting after brief instruction may prove to be an effective solution for patients facing various obstacles due to their inability to participate in therapy due to their place of residence or low economic status.

For RAS in post-stroke rehabilitation, we found that a programme involving GTBR (6 weeks), compared with standard gait training, was more effective for SP gait ability, balance and symmetry and lower limb function [18]. Moreover, when considering interventions to improve gait symmetry, it is worth considering the possible benefits of using GTBR beat frequency matching for fast step time. The researchers suggest that planned further research should focus on developing a method to use RAS in SPs for step time and step length. GTBR can be considered a useful method for functional rehabilitation in SPs. In addition, it may be applied in the home setting for outpatients. The reports analysed show that RAS-TT can contribute to the optimisation of gait rehabilitation in patients with SPs [19]. A significant improvement in gait velocity (*p* = 0.032) and cadence (*p* = 0.002) in the FGS was observed. The material analysed provides a starting point for inferring the greater effectiveness of RAS-TT over other approaches (i.e., TT and NDT) in functional gait rehabilitation and supports the development of a more optimal therapy by combining functional music with treadmill training. 

Results from the current literature also indicate that RAS is beneficial for improving gait speed and overall gait quality, stride length, or cadence in SPs [95,101]. However, the researchers point out that these reports should be treated with caution due to the high risk of bias in most reports. In the case of our study, we also faced uncertainties caused by missing information regarding blinding, deviations from the intervention, and missing results. Evidence from randomized studies with large samples is still lacking [101].

Music-based interventions have shown high efficacy in the treatment of neurological and neuropsychiatric disorders associated with stroke [102]. However, they are not only limited to motor function, but also language and cognitive function, as well as quality of life. Other studies show that music therapy can effectively improve dysphagia, which is a serious problem in the post-stroke elderly [103,104]. Melodic intonation therapy in both SP and PD therapy is gaining increasing interest among therapists and researchers [21,105,106]. Unfortunately, strong evidence for its effectiveness is still lacking. Researchers point out, among other things, the problems of providing the intervention to people who have not received adequate specialized training (musicians/music therapists), the lack of appropriate assessment tools and under-sampling in clinical trials [106]. Therefore, it can be concluded that research in this area should continue with the inclusion of techniques focusing on the speech/language area as well. Schaffert and colleagues (2019) explored based on the available literature (review), investigated the relationships existing between sound and movement in the context of both sports training and physical rehabilitation. The data presented prove the impact of natural movement sounds, rhythmic auditory information and movement sonication on sports training and (re)learning [107]. Another review of music therapy in PD found that most of the studies analysed show a beneficial effect of music therapy on the non-pharmacological treatment of motor and non-motor symptoms and on the quality of life of people with PD. The research included in this review covered all interventions using music in PD rehabilitation, not just NMT. Therefore, it is only an indication of the effectiveness of music in the rehabilitation of this disease [108].

It is worth noting that NMT contains 20 standardized and clinically applied techniques, most of which are used in therapeutic practice in the rehabilitation of older people. However, only one of them has qualified for our review. For example, for behavioural and psychological symptoms of dementia (BPSD), music therapy has proven to be a more effective and economically preferable option for improvement compared with pharmacotherapy [109]. It was also reported to improve mood and reduce anxiety and depression [110]. Of the NMT techniques, many are applicable to dementia therapy, so it is surprising that we have not found studies focusing on this subject. However, this is quite a young approach that is still gaining in popularity, and further research in this area is necessary.

The ability to observe the brain while processing musical information has contributed to a better understanding of how music works in therapy and, in effect, to develop a more precise scientific basis for music therapy [111]. Clinical studies indicate that the processes in the brain triggered by music can be generalized and transferred to non-musical functions, giving desired and measurable therapeutic effects [7,97,111,112]. NMT techniques are used in neurorehabilitation as non-invasive and promising results that should be considered as a complement to conventional cognitive neurorehabilitation and stimulation therapy [113]. Studies included in this review confirm this statement.

## 5. Conclusions

Due to the global population ageing worldwide and the need to integrate psychogeriatric disorders therapy, there is a growing need for effective non-pharmacological treatment and support for rehabilitation in the broadest sense of the term. Interventions using the RAS technique indicate that it is effective in rehabilitation of gait and fall reduction in PD patients and improve mobility in SPs. The combination of multimodal balance training with RAS and treadmill training with RAS was evaluated as effective intervention that helps optimize rehabilitation. The combination of auditory cues and music in the therapeutic programme seems to be beneficial for geriatric patients in terms of mobility, balance, falls and gait parameters (stride length, cadence, speed). NMT is an effective, non-invasive, low-cost, and accessible intervention for elderly patients; therefore, it is worth considering among non-pharmacological strategies to support sensorimotor rehabilitation in patients with PD and SP. The review indicates that modifications of the RAS are effective and also allow a better adaptation to the individual needs and abilities of the patient. Particularly as this technique can be applied at home settings.

RCTs are mainly conducted in the sensorimotor area and tend to focus on the RAS technique. However, NMT is still a niche therapeutic intervention, and it is worth focusing on further research on the use of RAS in sensorimotor rehabilitation and other NMT techniques in older patients, particularly in the areas of cognition and speech under different conditions.

## 6. Limitations

The restrictions we encountered during the creation of the review mostly concerned the nomenclature in the article abstract. The lack of systematic nomenclature hindered the process of selecting articles and raised several questions regarding the fulfilment of the inclusion criteria. A broad understanding of the concept of music therapy and a lack of precision in the methodology of conducting research on its effectiveness made some of the research have low reliability, and there was an inability to compare the latest discoveries with already available knowledge.

## Figures and Tables

**Figure 1 healthcare-10-02187-f001:**
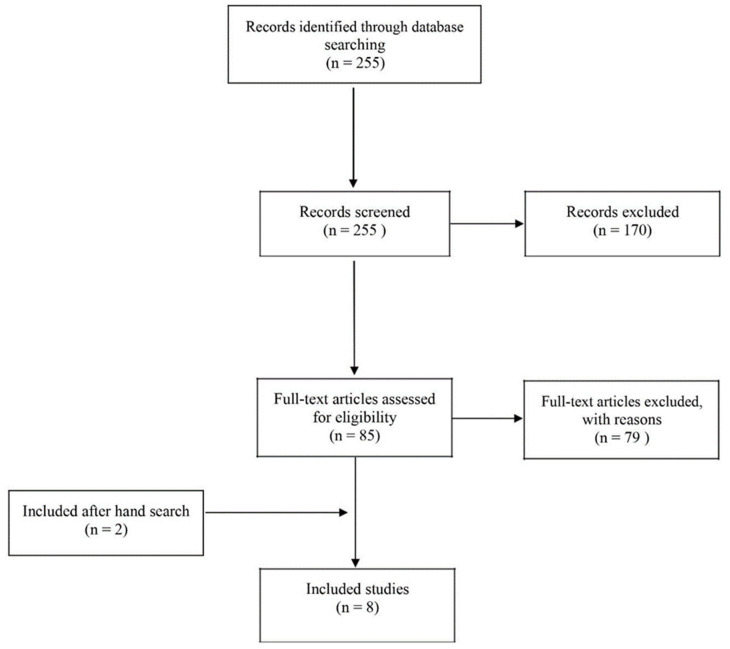
Flow chart of systematic review process.

**Figure 2 healthcare-10-02187-f002:**
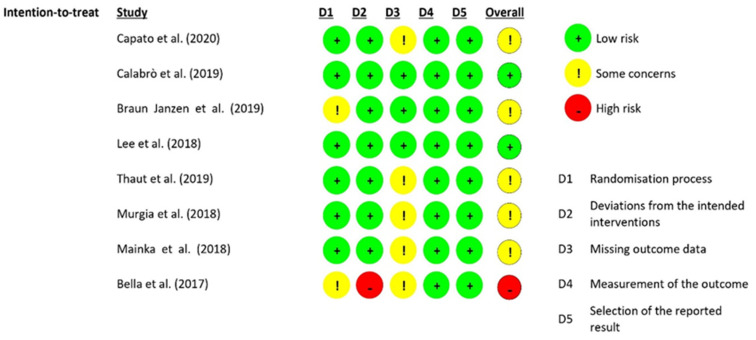
Risk of bias in included studies [12,13,14,15,16,17,18,19].

**Table 1 healthcare-10-02187-t001:** The division of neurologic music therapy techniques into three main groups that rehabilitate certain areas and the specific techniques used in them.

Domain	Name of Individual NMT Techniques
SENSORIMOTOR	Rhythmic Auditory Stimulation—RAS
	Patterned Sensory Enhancement—PSE
	Therapeutical Instrumental Music Performance—TIMP
SPEECH/LANGUAGE	Melodic Intonation Therapy—MIT
	Musical Speech Stimulation—MUSTIM
	Rhythmic Speech Cueing—RSC
	Vocal Intonation Therapy—VIT
	Therapeutic Singing—TS
	Oral Motor and Respiratory Exercises—OMREX
	Developmental Speech and Language Training Through Music—DSLM
	Symbolic Communication Training through Music—SYCOM
COGNITIVE	Musical Sensory Orientation Training—MSOT
	Musical Neglect Training—MNT
	Auditory Perception Training—APT
	Musical Attention Control Training—MACT
	Musical Mnemonic Training—MMT
	Associative Mood and Memory Training—AMMT
	Musical Executive Functions Training—MEFT
	Music in Psychosocial Training and Counselling—MPC
	Musical Echoic Memory Training—MEM

## Data Availability

The data presented in this study are available in Appendix A.

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
