# Peer review of "Neurologic Music Therapy in Geriatric Rehabilitation: A Systematic Review"

_healthcare, 2022, doi:10.3390/healthcare10112187_

Round 1
Reviewer 1 Report (Previous Reviewer 2)
Your paper does not have coherence from starting to finishing
I don't know what do you want to transmit in your review
.Why to describe in all detail the risk bias in all 8 studies and then analyze all of them again?
You have started with the wrong title and your introduction is confusing
As I suggest to the editors I may not be an adequate reviewer of your paper.
Author Response
SUMMARY OF OUR RESPONSES
Thank you for your careful review of our paper as well as your comments, corrections and suggestions that ensued. A careful revision of the paper has been carried out to take all of them into account, and in the process, we believe the paper has been significantly improved. In the present „Response Letter“ we first detail the major changes that have been made in the paper to correct the main weaknesses identified by the Reviewers. We then sequentially address all the points that we have corrected „step-by-step“. Changes that have been made to the text are highlighted in the 'track changes' mode.
The main changes:
- A Risk of Bias reassessment was conducted using the RoB - 2 tool,
- Table 2 has been removed from the main text,
- Risk of Bias report has been placed in supplementary materials,
- The results section was divided into smaller sections organizing the various RAS modifications,
- The discussion has been rewritten; the literature has been updated with publications from the last three years.
REVIEWER 1 EVALUATION
Your paper does not have coherence from starting to finishing
I don't know what do you want to transmit in your review
.Why to describe in all detail the risk bias in all 8 studies and then analyze all of them again?
You have started with the wrong title and your introduction is confusing
As I suggest to the editors I may not be an adequate reviewer of your paper.
Thank you for all your important comments. Our answers to your points are as follows:
Response: In our review we presented the results of literature research on specific NMT techniques used in psychogeriatrics. As we stated in the purpose of our review we tried to find answers to the following question: can NMT be effective in the rehabilitation of geriatric patients? Moreover, we wanted to locate areas where RCTs are being conducted. We decided on such a broad title because of the search strategy we followed. In the supplementary materials the full search strategy can be found, which includes both individual NMT techniques, tests outcomes from all three areas (sensorimotor, speech and cognitive) and applications to a variety of conditions/disorders. Our work has now been revised in line with the reviewers' suggestions. We have changed the Risk of Bias assessment tool and removed the extensive table describing these results from the main text. The discussion has also been edited. We belive that the current version is more readable and understandable.
We give full credit to the editor.

Reviewer 2 Report (New Reviewer)
Neurologic Music Therapy in Geriatric Rehabilitation: A Systematic Review. This study aimed to critically appraise the available literature on the application of particular NMT techniques in the rehabilitation of geriatric disorders. The review is interesting and, in principle, well done, although its credibility and quality would be improved by the following additions. Firstly, I was interested in the keywords used in the search strategy. Was the same strategy used in all databases, were the search terms verified with Mesh terminology and why the names of scales and tests were used among the terms and not the NMT methods themselves and the terms defining the target population. Combining the third group with an AND conjunct may have affected the search results. How many records were obtained before removing duplicates, how many after? The question arises from the fact that the number given in the text and on the flow chart are different. Why was it decided to use only the names of the techniques defined within the NMT by Thaut, and not to include at least the popular dance therapy in these patient groups?
Further concerns relate to the methodological assessment of the included primary studies presented. Indeed, the version published and recommended in the Cochrane guidelines was not used: ROB -2 for RCTs, even though the authors stated in the methods that they performed the review according to the Cochrane and PRISMA guidelines. Using the correct version, i.e. the ROB-2 recommended from 2019, would have affected the described risk of bias, as, for example, a low-risk value would not have been assigned for a study in which up to 7 of 35 subjects were lost to follow-up. I therefore recommend a reassessment, with the correct tool.
The discussion also needs to be rewritten. In it, the authors have restated the results and intervention methodology (RMS) of the primary studies rather than discussing them considering the results of other studies and reviews. This information only comes at the end of the discussion and should form its core. If RMS appeared to be the predominantly used technique, then the results could have summarized its various modifications, described more extensively in the primary studies, which would have provided a valuable summary for practitioners and therapists.
Author Response
SUMMARY OF OUR RESPONSES
Thank you for your careful review of our paper as well as your comments, corrections and suggestions that ensued. A careful revision of the paper has been carried out to take all of them into account, and in the process, we believe the paper has been significantly improved. In the present „Response Letter“ we first detail the major changes that have been made in the paper to correct the main weaknesses identified by the Reviewers. We then sequentially address all the points that we have corrected „step-by-step“. Changes that have been made to the text are highlighted in the 'track changes' mode.
The main changes:
- A Risk of Bias reassessment was conducted using the RoB - 2 tool,
- Table 2 has been removed from the main text,
- Risk of Bias report has been placed in supplementary materials,
- The results section was divided into smaller sections organizing the various RAS modifications,
- The discussion has been rewritten; the literature has been updated with publications from the last three years.
REVIEWER 2 EVALUATION
Thank you for all your important comments. Our answers to your points are as follows:
Neurologic Music Therapy in Geriatric Rehabilitation: A Systematic Review. This study aimed to critically appraise the available literature on the application of particular NMT techniques in the rehabilitation of geriatric disorders. The review is interesting and, in principle, well done, although its credibility and quality would be improved by the following additions. Firstly, I was interested in the keywords used in the search strategy. Was the same strategy used in all databases, were the search terms verified with Mesh terminology and why the names of scales and tests were used among the terms and not the NMT methods themselves and the terms defining the target population. Combining the third group with an AND conjunct may have affected the search results. How many records were obtained before removing duplicates, how many after? The question arises from the fact that the number given in the text and on the flow chart are different. Why was it decided to use only the names of the techniques defined within the NMT by Thaut, and not to include at least the popular dance therapy in these patient groups?
Response: Thank you for this valid comment. The full search strategy is given in the supplementary materials (PICO appendix). We have added an appropriate reference in the text so that there is no doubt about the search phase. We hope that this will dispel any concerns.
Another issue noted by the reviewer is due to our mistake. The number given in the text and in the flowchart are different due to an error in the generation of the diagram. We have replaced the previous figure with the correct version of the diagram. Thank you very much for your accurate observation.
The decision to use only the names of the techniques defined within the NMT was at the core of the planning of the study. We wanted to focus only on NMT techniques and the results they allow in RCTs in the field of geriatrics because of our professional experience in the practice area and our awareness of the therapeutic potential of individual techniques. We expected to find more studies testing techniques in the cognitive and sensorimotor area. We also did not include other methods, models, or approaches in music therapy. In the case of dance therapy, we would also have to distinguish very carefully between dance therapy and Dance Movement Therapy. In this case, we did not choose to broaden our search to include other art therapy methods and decided to stay only with the NMT model.
Further concerns relate to the methodological assessment of the included primary studies presented. Indeed, the version published and recommended in the Cochrane guidelines was not used: ROB -2 for RCTs, even though the authors stated in the methods that they performed the review according to the Cochrane and PRISMA guidelines. Using the correct version, i.e. the ROB-2 recommended from 2019, would have affected the described risk of bias, as, for example, a low-risk value would not have been assigned for a study in which up to 7 of 35 subjects were lost to follow-up. I therefore recommend a reassessment, with the correct tool.
Response: Thank you for this comment. We have decided to reassess Risk of bias using the RoB 2 tool. Due to this change, Table 2 and the figure 2 in the supplementary materials have been revised. A detailed report is available in the supplementary materials, while Figure 2 summarizing the results of the risk of bias assessment is included in the text. Certainly, this suggestion has improved our work, for which we are grateful.
The discussion also needs to be rewritten. In it, the authors have restated the results and intervention methodology (RMS) of the primary studies rather than discussing them considering the results of other studies and reviews. This information only comes at the end of the discussion and should form its core. If RMS appeared to be the predominantly used technique, then the results could have summarized its various modifications, described more extensively in the primary studies, which would have provided a valuable summary for practitioners and therapists.
Response: Thank you for this suggestion. We agree with it, so the discussion section has been rewritten to focus more on valuable findings and tips for practitioners and therapists, and has been supplemented with current literature. We hope that in this version it is more interesting, greatly improves the perception of the article, and makes a valuable contribution to practice.

Reviewer 3 Report (New Reviewer)
The authors should improve:
1. The wording of the objective should be improved. It is not possible to evaluate all the available literature but only that related to certain keywords that are not specified.
2. The presentation of the information in table 2. This table occupies almost 3 pages and therefore the difference between the different studies analyzed is not understood.
3. Section 3.3. should be systematized in relation to the studies. The type of studies should be defined: Parkinson's, music therapy techniques, etc.
4. The conclusions section is too brief
5. The references section should be expanded with current references 2022, 2021, and 2020. For example:
Abe, M., Tabei, K. I., & Satoh, M. (2022). The Assessments of Music Therapy for Dementia Based on the Cochrane Review. Dementia and Geriatric Cognitive Disorders Extra, 12(1), 6-1
Gassner, L., Geretsegger, M., & Mayer-Ferbas, J. (2022). Effectiveness of music therapy for autism spectrum disorder, dementia, depression, insomnia and schizophrenia: update of systematic reviews. European journal of public health, 32(1), 27-34.
Prieto Álvarez, L. (2022). Neurologic Music Therapy with a Habilitative Approach for Older Adults with Dementia: A Feasibility Study. Music Therapy Perspectives, 40(1), 76-83.
Author Response
SUMMARY OF OUR RESPONSES
Thank you for your careful review of our paper as well as your comments, corrections and suggestions that ensued. A careful revision of the paper has been carried out to take all of them into account, and in the process, we believe the paper has been significantly improved. In the present „Response Letter“ we first detail the major changes that have been made in the paper to correct the main weaknesses identified by the Reviewers. We then sequentially address all the points that we have corrected „step-by-step“. Changes that have been made to the text are highlighted in the 'track changes' mode.
The main changes:
- A Risk of Bias reassessment was conducted using the RoB - 2 tool,
- Table 2 has been removed from the main text,
- Risk of Bias report has been placed in supplementary materials,
- The results section was divided into smaller sections organizing the various RAS modifications,
- The discussion has been rewritten; the literature has been updated with publications from the last three years.
REVIEWER 3 EVALUATION
Thank you for all your important comments. Our answers to your points are as follows:
The authors should improve:
- The wording of the objective should be improved. It is not possible to evaluate all the available literature but only that related to certain keywords that are not specified.
Response: Thank you for this valid comment. The full search strategy is given in the supplementary materials (PICO appendix). We have added an appropriate reference in the text so that there is no doubt about the search phase. We hope that this will dispel any concerns.
- The presentation of the information in table 2. This table occupies almost 3 pages and therefore the difference between the different studies analyzed is not understood.
Response: Thank you for your comment. In order to improve the Risk of bias assessment in our study, we applied the RoB-2 tool and rephrased the text. Due to this change, Table 2 and the figure 2 in the supplementary materials have been revised. A detailed report is available in the supplementary materials, while Figure 2 summarizing the results of the risk of bias assessment is included in the text. We hope that the current version is more readable and fully understood by the reader.
- Section 3.3. should be systematized in relation to the studies. The type of studies should be defined: Parkinson's, music therapy techniques, etc.
Response: The results section has been divided into several subsections to facilitate understanding of the content presented. Section 3.1 included a brief description of the literature search, and Section 3.2 included information about the study groups and study types. Because only studies focused on the senomotor technique of RAS in two groups of patients met the inclusion criteria, we decided to create subsections 3.3 RAS in PD and 3.4 RAS in stroke. In this way, we systematized the studies by NMT technique and conditions. Title 3.3 has been expanded as "RAS in Parkinson's disease." Both subsections 3.3 and 3.4 have been subdivided into even smaller sections that clearly indicate which modification or combination of RAS the study involves and under what conditions.
- The conclusions section is too brief
Response: Thank you for this suggestion, due to the numerous amendments, this section was enriched.
- The references section should be expanded with current references 2022, 2021, and 2020. For example:
Abe, M., Tabei, K. I., & Satoh, M. (2022). The Assessments of Music Therapy for Dementia Based on the Cochrane Review. Dementia and Geriatric Cognitive Disorders Extra, 12(1), 6-1
Gassner, L., Geretsegger, M., & Mayer-Ferbas, J. (2022). Effectiveness of music therapy for autism spectrum disorder, dementia, depression, insomnia and schizophrenia: update of systematic reviews. European journal of public health, 32(1), 27-34.
Prieto Álvarez, L. (2022). Neurologic Music Therapy with a Habilitative Approach for Older Adults with Dementia: A Feasibility Study. Music Therapy Perspectives, 40(1), 76-83.
Response: Thank you for your attention. The discussion has been supplemented with current literature and rewritten to provide a valuable summary for practitioners and therapists. We trust that this suggestion has made our work more valuable.

Round 2
Reviewer 2 Report (New Reviewer)
Thank you very much for revising the manuscript according to my suggestions. I believe that a slightly broader framing of music therapy at the publication search stage (not limiting it only to the NMT definition) would have resulted in more publications included in the review, and thus the possibility of summarizing them in the meta-analysis. Such a framing of the review topic would be of value to clinicians involved in the rehabilitation of the elderly and would affect the readership of publications. I therefore leave it to the editor to decide whether the publication meets the requirements and expectations of the journal's readers. Substantively in its current state, it does not raise any objections from me, and I appreciate the contribution of the work and the fair response to the first review.
This manuscript is a resubmission of an earlier submission. The following is a list of the peer review reports and author responses from that submission.
Round 1
Reviewer 1 Report
The authors of a reviewed manuscript titled "Neurologic Music Therapy in Psychogeriatric Rehabilitation: A Systematic Review" looked at NMT techniques from an interesting angle, taking into consideration music interventions used for the psychogeriatric population. The provided systematic review was performed adequately. Only some minor errors were found, all of them are listed below.
Verse 68: the wrong name of the model was used. It is Transformation Project Model, it should be Transformation Design Model (TDM).
Verse 75: the aim of the review was formulated very ascetic. No specific research questions were established. It would be more interesting if the aim are more specifically described.
Paragraph (verses 77-89): The authors nothing mentioned about the verification process based on PRISMA. Was it used for selecting materials for the systematic review? It should be mentioned in this paragraph and the references should be added.
Paragraph (verses 102-112): There is no link to Figure 2 (supplement) in the text. I also wonder why the authors had not decided to use PEDro scale, which is commonly utilized in the systematic review for potential bias assessment and to evaluate the quality of the included studies. It is recommended to add PEDro scale to this study.
Paragraph (verses 403-409): The conclusions are slightly disappointing and treated minimalistically. The readers are waiting for some recommendations according to the most effective NMT techniques used for elderlies with psychogeriatric conditions.
Author Response
SUMMARY OF OUR RESPONSES
Thank you for your careful review of our paper as well as your comments, corrections and suggestions that ensued. A careful revision of the paper has been carried out to take all of them into account, and in the process, we believe the paper has been significantly improved. In the present „Response Letter“ we first detail the major changes that have been made in the paper to correct the main weaknesses identified by the Reviewer. We then sequentially address all the points that we have corrected „step-by-step“. Changes that have been made to the text are highlighted in the 'track changes' mode.
The main changes:
- The content of the article has been significantly shortened; non-essential content that hindered the reception of the paper has been removed,
- The aim of the study was expanded to include additional research questions to deepen understanding of the subject,
- We have clearly divided Table 3 (Participants of study groups in the literature review) according to the conditions of the subjects,
- Table 4 has been removed from the main text and transferred to supplementary material,
- The results section has been abbreviated,
- The summary of the review has been modified and extended and recommendations for the use of the RAS technique in the rehabilitation of patients with PD and SPs.
REVIEWER 1 EVALUATION
The authors of a reviewed manuscript titled "Neurologic Music Therapy in Psychogeriatric Rehabilitation: A Systematic Review" looked at NMT techniques from an interesting angle, taking into consideration music interventions used for the psychogeriatric population. The provided systematic review was performed adequately. Only some minor errors were found, all of them are listed below.
Thank you for all your important comments. Our answers to your points are as follows:
Verse 68: the wrong name of the model was used. It is Transformation Project Model, it should be Transformation Design Model (TDM).
Response: Thank you for this comment. The name of the Transformation Design Model has been corrected.
Verse 75: the aim of the review was formulated very ascetic. No specific research questions were established. It would be more interesting if the aim are more specifically described.
Response: We agree with this suggestion. Deepening the formulated objective of the literature review with additional research questions and hypotheses will certainly help to better understand our work and the subject matter presented. We have expanded the paragraph to read as follows: We tried to find answers to the questions: can NMT, as a relatively low-cost and safe method of influence, be effective in the rehabilitation of psychogeriatric patients? Which techniques, in which diseases and disorders are successfully applied? In addition, we wanted to locate areas (therapy and rehabilitation of individual conditions) where RCTs - providing strong evidence of effectiveness - are being conducted.
Paragraph (verses 77-89): The authors nothing mentioned about the verification process based on PRISMA. Was it used for selecting materials for the systematic review? It should be mentioned in this paragraph and the references should be added.
Response: Thank you for this valid comment. The verification system was based on the PRISMA guide to ensure transparent and complete reporting of our review. We have added relevant information and references.
Paragraph (verses 102-112): There is no link to Figure 2 (supplement) in the text. I also wonder why the authors had not decided to use PEDro scale, which is commonly utilized in the systematic review for potential bias assessment and to evaluate the quality of the included studies. It is recommended to add PEDro scale to this study.
Response: Thank you for this comment. We have added a link to Figure 2 attached in the supplementary material (lines 112/113). In our study, the assessment of potential bias was performed according to The Cochrane Collaboration's tool for assessing risk of bias. Reference to the source can be found in subsection 2.3 Assessment of risk of bias (lines 113-114). It is validated method for risk of bias assessment.
Sources:
- Cochrane Handbook for Systematic Reviews of Interventions; Higgins, J.P.T., Thomas, J., Chandler, J., Cumpston, M., Li, T., Page, M.J., Welch, V.A., Eds.; 1st ed.; Wiley, 2019; ISBN 978-1-119-53662-8.
- https://handbook- 5- 1.cochrane.org/chapter_8/table_8_5_d_criteria_for_judging_risk_of_bias_in_the_risk_of.htm (online)
- Website: https://training.cochrane.org/handbook/archive/v6.2/chapter-08
Paragraph (verses 403-409): The conclusions are slightly disappointing and treated minimalistically. The readers are waiting for some recommendations according to the most effective NMT techniques used for elderlies with psychogeriatric conditions.
Response: We agree with this suggestion. The summary has been modified and expanded to include recommendations and comments on both the effectiveness of the application of the RAS technique in specific conditions and on the directions in which further research should be conducted.
We hope that the changes made have significantly improved the quality of our work. Once again, we would like to thank you for your insightful review and suggestions that have helped us in this process.

Reviewer 2 Report
I have difficulties with this paper from the titles down. Needs a full new construction.You cannot compare PD with stroke patients :neurodegenerative with vascular pathology?Even if the RAS is a very good technique, cannot be compared in these samples.
Unfortunately there is too much information in descriptions ,and make the content difficult to understand for the readers.Suggestions:
*Introduction: there are 52initial lines of unnecessary info.
*Too long the info about Risk of bias>Is distracting
* Results must be abbreviated and the analysis of the different paper are unnecessary.
* Table 3 putting together(again) PD and strokes is mixing different subject/themes.
* table 4 is another repetitive information
* authors should go back to the basic idea of the application of NMT in a specific condition, and start from there. Psychogeriatric is not a good starting point ,is distracting. But go back:you already did a good job looking for the available data.
Author Response
SUMMARY OF OUR RESPONSES
Thank you for your careful review of our paper as well as your comments, corrections and suggestions that ensued. A careful revision of the paper has been carried out to take all of them into account, and in the process, we believe the paper has been significantly improved. In the present „Response Letter“ we first detail the major changes that have been made in the paper to correct the main weaknesses identified by the Reviewer. We then sequentially address all the points that we have corrected „step-by-step“. Changes that have been made to the text are highlighted in the 'track changes' mode.
The main changes:
- The content of the article has been significantly shortened; non-essential content that hindered the reception of the paper has been removed,
- The aim of the study was expanded to include additional research questions to deepen understanding of the subject,
- We have clearly divided Table 3 (Participants of study groups in the literature review) according to the conditions of the subjects,
- Table 4 has been removed from the main text and transferred to supplementary material,
- The results section has been abbreviated,
- The summary of the review has been modified and extended and recommendations for the use of the RAS technique in the rehabilitation of patients with PD and SPs.
REVIEWER 2 EVALUATION
I have difficulties with this paper from the titles down. Needs a full new construction.You cannot compare PD with stroke patients :neurodegenerative with vascular pathology? Even if the RAS is a very good technique, cannot be compared in these samples.
Unfortunately there is too much information in descriptions ,and make the content difficult to understand for the readers. Suggestions:
Thank you for all your important comments. Our answers to your points are as follows:
*Introduction: there are 52initial lines of unnecessary info.
Response: Thank you for this comment. The popularity of using non-pharmacological therapies, including music therapy, is growing every year. Unfortunately, available research shows that music therapy is often equated with the use of music to achieve specific goals. It is difficult to find information regarding the leading of an intervention by a qualified professional; the stream or model on which the research is based, as well as the methods or techniques. The extended introduction was intended to introduce the reader step by step to the subject. We understand that too much information makes it difficult to receive the article, so we decided to shorten the introduction, in line with the reviewer's comment. We have marked the deleted passages in the text by strikethrough.
*Too long the info about Risk of bias>Is distracting
Response: In line with the reviewer's comment, we have shortened the subsection 'Assessment of risk of bias', leaving two sentences covering the most important content and Table 2 entitled: "Risk of bias in included studies". We hope that this action will facilitate the reception of this section of our paper.
* Results must be abbreviated and the analysis of the different paper are unnecessary.
Response: In line with the reviewer's comment, we have decided to modify the results section. In order to abbreviate, we have removed Table 4 from the main text. Due to its usefulness (summary of all articles included in the study), it will be available in the supplementary materials. In our opinion, a brief description of each paper is important for the reader due to the lack of possibility to compare papers with each other. In line with the reviewer's comment, we have abbreviated the results presented. In order to make the results clear and not mislead the reader, we have presented the studies on Parkinson's disease and stroke in separate subsections.
* Table 3 putting together(again) PD and strokes is mixing different subject/themes.
Response: Thank you for this comment. We have decided to modify the table and clearly separate the RAS technique for each condition in Table 3, as well as listing the same types of disorder side by side.
* table 4 is another repetitive information
Response: Thank you for your comment. We believe that reducing the paper according to the reviewer's comments contributes to its clarity and improves its quality. Table 4 has been removed from the main text, as we mentioned above.
* authors should go back to the basic idea of the application of NMT in a specific condition, and start from there. Psychogeriatric is not a good starting point ,is distracting. But go back:you already did a good job looking for the available data.
Response: Thank you for your insightful comments, which undoubtedly serve to improve the quality of our article. Of course, we are not able to compare a disorder of a neurodegenerative nature with a disorder of vascular pathology (this was not our intention). Instead of a comparison, we have chosen to present the results, in separate sections, of the use of NMT in individual conditions. Also, the modifications of the RAS training do not allow an exact comparison of the effects within the groups. When starting the study, we hoped to be able to locate and present areas where randomised controlled trials on the efficacy of NMT in older patients are being conducted. We expected to identify more studies incorporating cognitive techniques in patients with dementia, or speech therapy techniques after strokes, for example. Even the absence of the other two techniques (TIMP and PSE) was a surprise to us, as experience from clinical practice shows that these techniques are used in psychogeriatrics. Unfortunately, also the papers we rejected show a lack of understanding of the use of professional music therapy measures in psychogeriatrics. Nevertheless, our review presents our findings and provides some basis that will perhaps encourage readers to continue their search for strong evidence of the effectiveness of NMT techniques, we hope.
In our view, our review shows research gaps that need to be filled. NMT consists of standardised techniques that are compatible with clinical application. Although trends are emerging worldwide for the use of safe and non-pharmacological forms of therapy, there is still a lack of strong arguments for the dissemination of this model. We chose to explore the field of psychogeriatrics because of the global ageing population and the challenges that the public health field is already facing. For us as professionals working with older people, this is an extremely important topic. There are currently 38 million people in Poland, 16% of whom are over 65 - and there are 5 qualified neuro-music therapists (data from The Academy of Neurologic Music Therapy database). We are facing a lack of financial resources, inefficiencies in the primary healthcare system and an ever-increasing number of psychogeriatric patients. This is certainly a general problem. We need systemic change and greater dissemination of various non-pharmacological forms of therapy that can support rehabilitation, are effective and benefit patients in different areas.
We are therefore keen to share the results of our work and believe that they can provide a basis, a first step for further inquiries in this field.
We hope that the changes made have significantly improved the quality of our work. Once again, we would like to thank you for your insightful review and suggestions that have helped us in this process.

Round 2
Reviewer 2 Report
I have problems with this review from the title on!! Psychogeriatrics is referred to as Psychological, and Psychiatric disorders/diseases in the elderly!!
Then there is no coherence in the text throughout the paper
Of the final 8 studies chosen, 4 showed concern??
Unnecessary tedious information regarding the "risk of bias"
Where is the data of the effectiveness of the NMT/RAS intervention? where is the significance?
This Review --as is-- does not show any advantage in using the technique in both Neurodegenerative and Vascular pathology